# DeepMMSearch-R1: Empowering Multimodal LLMs in Multimodal Web Search

## Abstract

Multimodal Large Language Models (MLLMs) in real-world applications require access to external knowledge sources and must remain responsive to the dynamic and ever-changing real-world information in order to address information-seeking and knowledge-intensive user queries. Existing approaches, such as retrieval augmented generation (RAG) methods, search agents, and search equipped MLLMs, often suffer from rigid pipelines, excessive search calls, and poorly constructed search queries, which result in inefficiencies and suboptimal outcomes. To address these limitations, we present **DeepMMSearch-R1**, the first multimodal LLM capable of performing on-demand, multi-turn web searches and dynamically crafting queries for both image and text search tools. Specifically, DeepMMSearch-R1 can initiate web searches based on relevant crops of the input image making the image search more effective, and can iteratively adapt text search queries based on retrieved information, thereby enabling self-reflection and self-correction. Our approach relies on a two-stage training pipeline: a cold start supervised fine-tuning phase followed by an online reinforcement learning optimization. For training, we introduce MMWebSearchVQA, a novel multimodal VQA dataset created through an automated pipeline intermixed with real-world information from web search tools. This dataset contains diverse, multi-hop queries that integrate textual and visual information, teaching the model when to search, what to search for, which search tool to use and how to reason over the retrieved information. We conduct extensive experiments across a range of knowledge-intensive benchmarks to demonstrate the superiority of our approach. Finally, we analyze the results and provide insights that are valuable for advancing multimodal web-search.

## 1 Introduction

Multimodal Large Language Models (MLLMs) (Hurst et al., 2024; Team et al., 2023; Li et al., 2024a; Bai et al., 2025; Chen et al., 2024; You et al., 2023; Wang et al., 2024; Deitke et al., 2024) combine pre-trained visual encoders with large language models (LLMs), and have achieved remarkable progress across a range of visual perception, grounding and generation tasks. These capabilities have made them integral to a wide range of everyday intelligent assistance applications. Despite these advances, they continue to struggle with knowledge-intensive and information-seeking visual question answering (VQA) (Chen et al., 2023; Mensink et al., 2023), which requires not only accurate recognition of visual entities but also access to relevant background knowledge. The sheer breadth of open-world visual knowledge places many queries in the long-tail distribution, and inevitably demands information beyond a model's internal training corpus. Constructing ever-larger training datasets is impractical, as it requires costly pipelines of data collection, cleaning, organization, and retraining. Furthermore, because the web is continuously updated, static training corpora quickly become outdated, leaving MLLMs unable to answer questions that require access to up-to-date information. For example, Qwen2.5-VL (Bai et al., 2025), last updated on January 2025, fails to

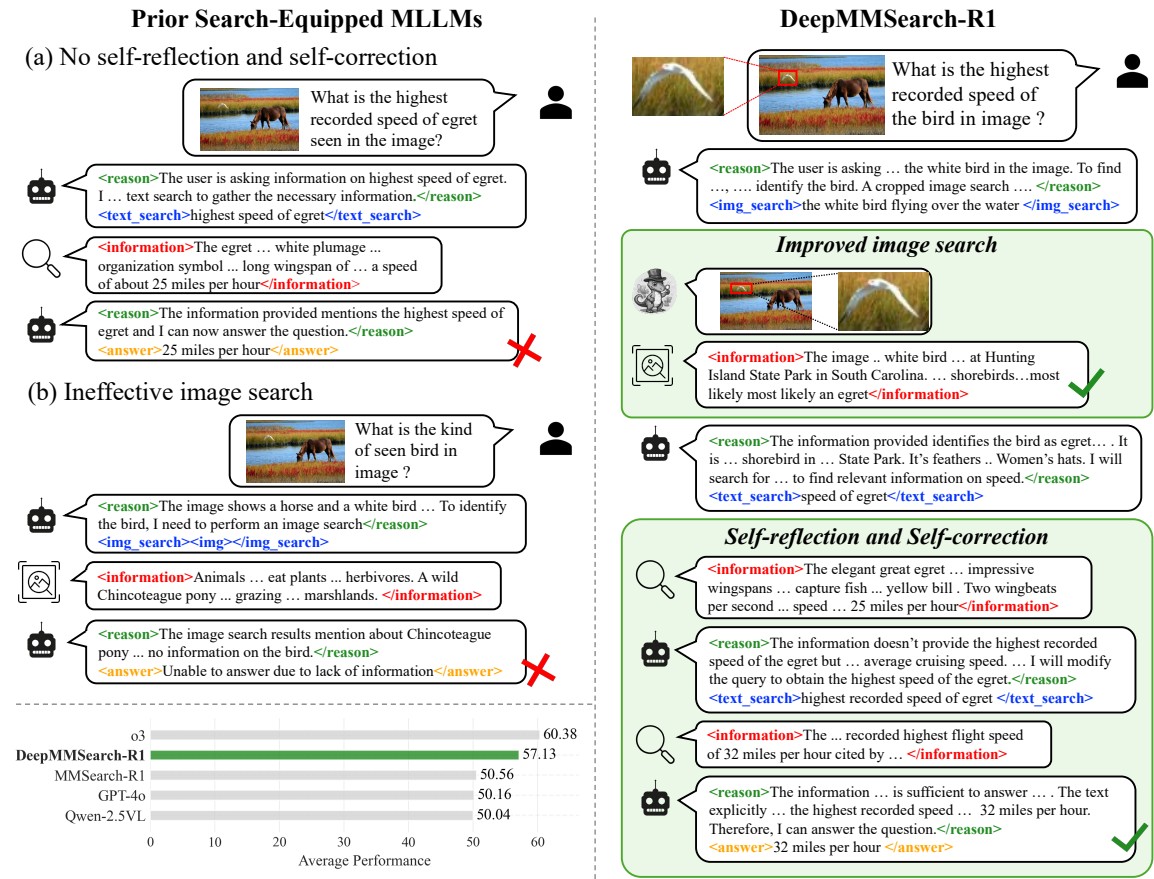

Figure 1: Unlike previous baselines, which lack self-reflection, self-correction, and cropped image-based search, the proposed **DeepMMSearch-R1** is capable of performing on-demand, multi-turn web searches with enhanced image search that incorporates an intermediate cropping tool to select the most relevant region of an image. It demonstrates self-reflection and self-correction abilities, iteratively refining its text queries to better navigate noisy real-world web information. The model outperforms other baselines, notably GPT-4o, and is competitive with the GPT-o3 model.

answer:"<image> even timage </image>Where is the boat race happening?" [Answer: The image shows the annual Pacu Jalur boat races in Riau Province, Indonesia]. The image is provided in Appendix F.1.

To address these limitations, recent research has sought to integrate search tools with MLLMs to provide dynamic access to external information. These existing approaches can be broadly classified into three categories: **(1) Retrieval-Augmented Generation (RAG) methods:** Ling et al. (2025); Qi et al. (2024); Liu et al. (2024e) which rely on external knowledge bases; however, no static corpus can capture the full breadth of open-world knowledge, making this assumption unrealistic in practice. Furthermore, the rigid retrieve-then-generate pipeline of RAG-based methods often results in excessive and unnecessary retrieval. **(2) Search Agents:** Li et al. (2024c;b) prompt LLMs/MLLMs to perform multi-turn web searches and incorporate the retrieved content into the model's context for subsequent turns. These agents are typically implemented as plug-and-play modules rather than being optimized for interaction with noisy, real-world web-search results. As a consequence, they often fail to reason effectively over retrieved content and struggle

to generalize in open-world scenarios not seen during pretraining. More recent efforts fall in the category of **(3) Search-Equipped MLLMs** (Jin et al., 2025; Song et al., 2025; Chen et al., 2025), which are trained to operate in unison with search tools and to reason over retrieved content. However, most existing works remain confined to text search, severely constraining their applicability to multimodal knowledge-intensive question answering. Wu et al. (2025) is the only work that extends retrieval into the multimodal domain by incorporating an image search tool. Nonetheless, it faces significant limitations. *First*, while the model can autonomously decide which tool to invoke, it is restricted to a single call per tool, limiting its capacity for self-reflection and self-correction. *Second*, in information-seeking and knowledge-intensive VQA tasks, accurately identifying the specific visual entity in the image that the question targets is crucial. In real-world deployment, however, background content and additional visual entities often introduce noise during image search. This noise can lead to suboptimal retrieval and incorrect identification of the relevant visual entity, creating a major bottleneck that renders image search largely inefficent in practice (see Figure 1 (left)).

To overcome the two key limitations identified in prior works, we propose **DeepMMSearch-R1**, a model capable of performing on-demand, multi-turn web searches with dynamic query generation for both image and text search tools as shown in Figure 1(right). Specifically, DeepMMSearch-R1 can adaptively generate and refine text-search queries over multiple turns through self-reflection and self-correction, using the retrieved content as feedback along with the original question. To improve the effectiveness of image search, we address the challenges posed by background noise and the presence of distracting visual entities by introducing an intermediate image cropping tool, which in our case is Grounding DINO (Liu et al., 2024b). DeepMMSearch-R1 first generates a concise description of the visual entity most pertinent to the question, which is then used by the cropping tool to dynamically identify and crop the corresponding region of the image. The resulting crop is used for image search, retrieving more contextually relevant results. This targeted approach significantly enhances retrieval quality and significantly boosts overall performance. We adopt a two-stage training pipeline consisting of an initial supervised fine-tuning (SFT) phase followed by online reinforcement learning (RL) using GRPO algorithm (Shao et al., 2024). Our goal is to teach the model *when to search*, *which tool to use*, *what to search for*, and *how to reason over retrieved content to determine the next action*, whether that is providing a final answer or refining the query for another search. Our main contributions are summarized below:

1. **Proposed Dataset:** We introduce *DeepMMSearchVQA*, a novel dataset containing diverse, multi-hop VQA samples with multi-turn conversations. It offers a balanced representation across knowledge categories and includes both search-required and search-free questions. The dataset teaches the model: when and what to search, which tool to use, and how to reason over retrieved content.

2. **Multimodal Search Tool Integration**: We construct a real-world multimodal search pipeline composed of three tools: (1) *a text search tool* that enables the model to issue targeted queries for retrieving relevant webpages and acquiring up-to-date factual knowledge; (2) *a grounding tool* (Grounding DINO Liu et al. (2024b)) that identifies and crops the relevant region of an input image based on a model-generated textual description of the visual entity in question; and (3) *an image search tool* that retrieves web content, including titles and descriptions, based on the input image (cropped or whole), helping the model gather web information to recognize unfamiliar visual entity.

3. **Performance Improvement:** We achieve state-of-the-art performance, surpassing previous open-source baselines (see Figure 1), through a two-stage training process: cold-start initialization with SFT, followed by online RL using GRPO. We discuss the impact of self-reflection, self-correction, and cropped image search, and provide additional analysis of tool call behavior, which together serve as a valuable resource for advancing multimodal web-search tool integration in MLLMs.

## 2 PROPOSED DATA: DEEPMMSEARCHVQA

There is a lack of instruction tuning dataset to equip multimodal LLMs with web-search capabilities. To fill this gap, we propose DeepMMSearchVQA, consisting of multi-turn conversations that integrate structured

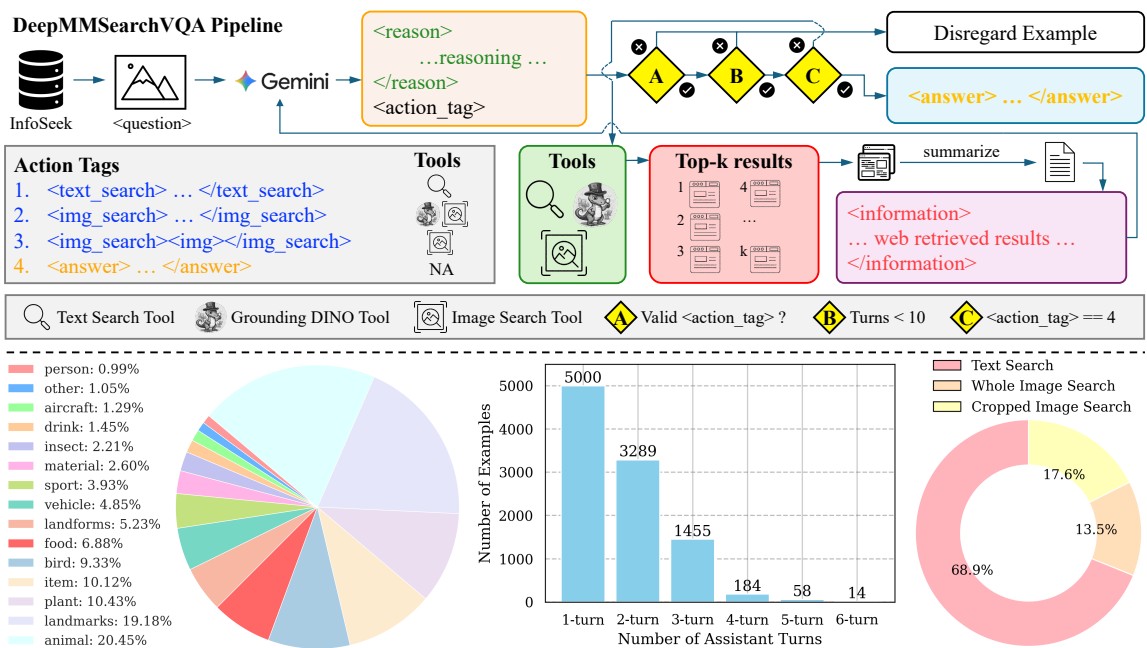

Figure 2: **(top)** *DeepMMSearchVQA Data Generation Pipeline:* It begins by passing a question–image pair $(q, i)$ to Gemini, which produces reasoning and concludes with an *action tag*. We then apply checks A, B, and C: if either A or B fails, the example is discarded; if C passes, the final answer is reached and the example is saved. Otherwise, the pipeline invokes a search tool guided by the action tag. This tool retrieves the top-$k$ web results, which are then summarized and fed back into Gemini, incorporating web-retrieved information in its context for subsequent turns in the reasoning process. **(bottom)** *DeepMMSearchVQA Statistics:* Knowledge taxonomy, Distribution of examples across different numbers of conversational turns, Proportion of questions with image search, text search and both tool calls.

tool call annotations and web-retrieved information obtained from both image and text search tools. We follow two core principles for the dataset generation: (1) the dataset should be diverse and cover the entire spectrum of the knowledge taxonomy, and (2) the questions should include both search-free and search-required types, with multiple conversational turns to foster reasoning, self-reflection and self-correction. An overview of the automated pipeline used for dataset construction is presented in Figure 2(top).

We employ the InfoSeek (Chen et al., 2023) train set as our base corpus and generate the conversations with reasoning distilled from Gemini-2.5-Pro Team et al. (2023). We provide the prompt used in Appendix E.1. The model decides which tool to invoke and what query to issue, outputting structured tool tags that are included in each training example. If the entire image is relevant to answering the question, the model appends `<img_search>img</img_search>` at the end of its output. When the question pertains to a specific visual entity within the image, such as an object, logo, or person, the model invokes a cropped image search query by appending `<img_search>[concise description of the visual entity]</img_search>`. In cases where the model can confidently identify the visual entity but requires additional factual information from external sources, it invokes the text-search tool with a focused query by outputting `<text_search>[search query]</text_search>`. In some examples, the model issues multiple refined text searches, which are crucial for capturing self-reflection and self-correction capabilities. Once sufficient information has been gathered, the model provides the final response inside as `<answer>[final answer]</answer>` tag. Before generating any tool tag, the model

explains its decision inside `<reason>...</reason>` block, ensuring that the reasoning can be captured in the dataset. All the information retrieved from image or text search is returned to the model within `<information>...</information>`, which is then incorporated into subsequent reasoning and tool selection. This structured interaction design allows us to capture Gemini-2.5-Pro's reasoning, tool selection, self-reflection, and self-correction capabilities in our training dataset. An illustration of the data is shown in Figure 1, and we provide more examples in Appendix G.

We randomly select a subset of 200,000 samples from the InfoSeek train set and generate multi-turn conversations annotated with tool tags, reasoning steps, and web-retrieved information. To ensure quality, we retain only those conversations in which Gemini-2.5-Pro's predictions match the ground-truth answers provided in InfoSeek, yielding a refined set of approximately $47,000$ conversations. Subsequently, we employ Gemini-2.5-Pro to categorize the questions according to a knowledge taxonomy. From these categories, we sample 10,000 VQA examples to achieve an approximately balanced distribution across knowledge types. We further ensure that the dataset is evenly divided between search-required and search-free questions. Figure 2(bottom) presents the knowledge taxonomy, the proportion of questions requiring image search, text search, or both, as well as the distribution of examples across different numbers of conversational turns. The resulting set of 10,000 VQA samples constitutes the training corpus for the supervised finetuning stage.

## 3 DEEPMMSEARCH-R1 TRAINING RECIPE

We follow a two-stage training pipeline. In the first stage, we perform supervised fine-tuning as a cold-start initialization. This equips the model with grounding, image search and text search tools, and enables it to reason over the web-retrieved content. In the second stage, we perform an online GRPO optimization to further refine the model's tool-selection ability and improve the efficiency of its search behavior.

### 3.1 SUPERVISED FINETUNING STAGE

We employ `Qwen2.5-VL-7B-Instruct` as our base model and perform supervised fine-tuning exclusively on the LLM module, while keeping both the vision encoder and vision projection layers frozen. This approach preserves the strong pretrained image representations and ensures that adaptation is directed toward enhancing the LLM's ability to reason over web-retrieved information and adhere to structured tool-usage protocols. To enable efficient training, we incorporate LoRA adapters with a rank of $r = 8$ across all transformer blocks of the LLM, thereby providing sufficient expressivity to capture the new behaviors required for web-information augmented reasoning while maintaining a manageable number of trainable parameters. The SFT data consists of multi-turn conversations that include reasoning sequences, tool-call annotations, and web-retrieved content from search tools. Through exposure to these structured conversations, the model learns when to initiate searches, which tool to use, how to formulate effective queries, how to integrate retrieved information into reasoning, and how to comply with the strict formatting conventions, which are all necessary for the seamless integration of search tools.

**Training Objective.** We adopt the standard Causal Language Modeling (Causal LM) objective. Given a multimodal input $(x, I)$, consisting of a textual question and an accompanying image, along with a multi-turn conversation $y^*$ that includes the complete reasoning trace, tool calls, and final answer, the model is trained to predict each token in the target sequence conditioned on all previous tokens:

$$\mathcal{L}_{\text{SFT}} = -\sum_{t=1}^{T} \log \pi_\theta(y_t^* \mid x, I, y_{<t}^*).$$

Here, $T$ denotes the length of the target sequence, and $\pi_\theta$ is the model's conditional distribution. Importantly, the web-retrieved information from the search tools are masked during loss computation, ensuring that training is concentrated on reasoning and structured tool calls, rather than being influenced by raw web-retrieved information.

## 3.2 REINFORCEMENT LEARNING STAGE

**GRPO.** The reinforcement learning stage relies on Group-Relative Policy Optimization (GRPO), introduced in DeepSeekMath (Shao et al., 2024). GRPO extends Proximal Policy Optimization (PPO) by stabilizing training through comparisons among candidate responses generated for the same prompt. Instead of evaluating each rollout independently, GRPO computes advantages relative to the mean reward within a group of sampled rollouts. Given an input $(x, I)$, the policy generates $K$ rollouts $\{y^{(i)}\}_{i=1}^K$, each associated with a reward $R^{(i)}$. The advantage for a single rollout is then defined as $A^{(i)} = R^{(i)} - \bar{R}$, where $\bar{R}$ is the average reward across the group. This centering removes the dependency on the absolute scale of rewards and focuses the optimization on identifying responses that are better than the group average. The objective is then optimized with a clipped importance-weighted surrogate, similar to PPO, but incorporating this group-relative advantage. Mathematically, the update is expressed as

$$\mathcal{L}_{\text{GRPO}} = \mathbb{E}_{i,t}\Big[ \min\big(\rho_t^{(i)} A^{(i)},\ \text{clip}(\rho_t^{(i)}, 1 - \epsilon, 1 + \epsilon) A^{(i)}\big)\Big] - \beta \, \text{KL}(\pi_\theta \,\|\, \pi_{\text{ref}}),$$

where $\rho_t^{(i)}$ is the ratio between the probabilities of a token under the current and old policies, $\epsilon$ controls the clipping range, and $\beta$ scales the KL regularization with respect to a frozen reference model. This formulation encourages relative improvements within each batch of responses, yielding stable optimization even under noisy reward signals.

**Rollouts.** The rollouts are generated from the model checkpoint after SFT. The SFT model interacts with the grounding tool, image search tool, and text search tool using the learned tool-call tag schema, incorporating feedback from these tools into subsequent turns. This process continues until either a final response is produced or the maximum number of turns is reached. When generating responses, if the model cannot confidently identify a visual entity in an image, it initiates either a full-image search or a cropped-image search. If the entity is identifiable but additional factual information is required, the model issues one or more text search queries to retrieve relevant details from the web. Each rollout thus represents a complete reasoning trajectory, annotated with the tag schema learned during SFT. During training, constraints are applied on the number of tool calls and the maximum token length per trajectory, requiring the model to balance accuracy with efficiency.

**Reward.** The GRPO optimization uses a composite reward balancing factual accuracy and structural compliance. We employ `gpt-5-chat-latest` as the reward model which judges semantic correctness of the predictions against the ground truth. The correctness score, denoted $s$, is binary ($s \in \{0, 1\}$), indicating whether the model's final answer is judged correct. In parallel, a format score $s_{\text{fmt}}$ measures adherence to the required output schema, ensuring correct tag usage and valid tool-call structure. The final reward is computed as $R_{\text{total}} = (1 - \lambda_{\text{fmt}}) s + \lambda_{\text{fmt}} s_{\text{fmt}}$, where $\lambda_{\text{fmt}}$ is the format penalty coefficient. This reward formulation drives the model to produce responses that are both factually reliable and consistent with the structured protocol required for seamless tool use.

## 4 EXPERIMENTS

### 4.1 EXPERIMENTAL SETUP

**Multimodal Search Tools:** To retrieve additional context and up-to-date information, we employ a multimodal search pipeline composed of three tools: a text search tool, an image search tool, and a grounding tool. The text search tool operates on DeepMMSearch-R1 generated textual queries, which are processed by an in-house web search API to retrieve relevant documents from a large-scale index. The top five results are then condensed by an LLM-based summarization module, producing concise outputs directly relevant to the user's question. To retrieve additional information about a visual entity, DeepMMSearch-R1 is trained to utilize the image search tool. When the model determines that the question concerns only a specific region of the image, it first produces a referring expression that concisely describes the region of interest. GroundingDINO (Liu et al., 2024b) is then employed to ground this expression, yielding a bounding box that is cropped and used as input for retrieval. In other cases, when the entire image is relevant, the original image is directly used without grounding. The resulting visual input is then passed to our in-house image search API, which retrieves visually similar images from the web along with surrounding context and page metadata. An LLM-based summarization module condenses the top five retrieved results, producing a concise description of the visual entity. Neither the text nor the image search index or API were modified for use with DeepMMSearch-R1, demonstrating the model's ability to operate seamlessly with standardized retrieval tooling. We employ `gpt-5-chat-latest` as the LLM summarizer to summarize search results of both the tools. This step is essential for keeping the retrieved content concise in order to avoid exceeding the model's maximum context length. The prompt used for summarization is provided in Appendix E.7

**Implementation Details:** We finetune `Qwen2.5-VL-7B-Instruct` using the LLaMA-Factory framework Zheng et al. (2024) with LoRA (rank 8) applied across all target modules. Training is performed for 3 epochs with a learning rate of $1e^{-4}$, cosine scheduler, and bf16 mixed precision on 1 node with 8 H100 GPUs. The global batch size is 8, and input masking is applied to optimize only on generated outputs for the multi-turn VQA dataset. For online RL optimization, we adopt the GRPO algorithm in the veRL Sheng et al. (2024) framework, using `gpt-5-chat-latest` as the reward model. We set $\lambda_{\text{fmt}} = 0.1$, apply a KL-penalty of 0.001, and use a clip ratio of 0.2. Training runs for 30 epochs on 4 nodes $\times$ 8H100 GPUs with a batch size of 512, and rollout number of 8. A warmup phase of 45 steps is applied with learning rate initialized at $2e^{-6}$. The maximum response length is 8192 tokens, and input masking is again used to focus optimization on generated outputs. Additional implementation details are provided in Appendix D.

**Baselines:** To benchmark the effectiveness of DeepMMSearch-R1, we evaluate it against a diverse set of strong baselines, including closed-source models (GPT-4o and GPT-o3) as well as open-source models from the Qwen2.5-VL family. We organize our comparisons into four evaluation workflows: (1) *Direct Answer*, where models are instructed to produce a concise answer without using any external retrieval; (2) *RAG Workflow*, where models are required to perform exactly two retrieval steps for each question, first conducting an image search, followed by a text search. In this setting, the retrieved image results are provided in the first round to guide text query generation, and the retrieved text results are supplied in the second round to produce the final answer. While this workflow maximizes exposure to external knowledge, it can also introduce noise when irrelevant or low-quality search content is retrieved; (3) *Prompt based Search Agents*, where the base model is prompted to make use of the multimodal search tools and the retrieved results are incorporated in generating the final response; and (4) *Web-search-equipped MLLMs*, which refers to models capable of performing on-demand, multi-turn search. Prior works such as Search-R1 (Jin et al., 2025) and ReSearch (Chen et al., 2025) are restricted to text-based retrieval and therefore cannot be considered true baselines. MMSearch-R1 (Wu et al., 2025), on the other hand, supports multimodal retrieval and serves as our only baseline. The prompts used for all the workflows are detailed in the Appendix E.

**Datasets:** We use the InfoSeek (Chen et al., 2023) dataset to construct DeepMMSearchVQA, which serves as the training set for SFT stage. For online GRPO optimization, we employ the FVQA (Wu et al., 2025) training set. We select the FVQA dataset because it contains a higher proportion of questions requiring image search compared to the InfoSeek dataset used in the SFT stage. This choice encourages more frequent image search tool calls, which is essential for achieving better performance in multimodal information-seeking VQA. For evaluation, we employ the InfoSeek test split along with DynVQA (Li et al., 2024c), SimpleVQA (Cheng et al., 2025), Encyclopedic-VQA (Mensink et al., 2023), OKVQA (Marino et al., 2019), and A-OKVQA (Schwenk et al., 2022) as benchmark datasets. Due to the large size of InfoSeek and Encyclopedic-VQA, we randomly sample 2000 examples from the test split of each. For SimpleVQA, we include all QA examples written in English. OK-VQA and A-OKVQA consist of relatively simple questions derived from COCO Lin et al. (2014) images, requiring little to no search. These benchmarks evaluate models on outside-knowledge questions, but since many modern MLLMs now include COCO in their pretraining Bai et al. (2025), the datasets have become easier and largely search-free.

**Evaluation Metric:** We evaluate model performance using the LLM-as-Judge framework, where a LLM is employed to assess the accuracy of responses. We adopt this approach to capture nuanced correctness in the multimodal, open-ended VQA task, which is often challenging for traditional automatic metrics. Specifically, we use `gpt-5-chat-latest` as the judging model. It is provided with the question, the ground-truth answer, and the model's response, and then determines whether the response is correct. The full evaluation prompt is provided in Appendix E.5.

## 4.2 RESULTS AND ANALYSIS

**Web-search equipped MLLMs outperform RAG workflows and prompt-based search agent baselines.** As shown in Table 1, DeepMMSearch-R1-7B (RL) surpasses both RAG workflows and prompt-based search agent baselines by a significant margin ($+21.13$ and $+8.89$ respectively), while achieving competitive performance with the OpenAI o3 model (OpenAI, 2025). On datasets such as OK-VQA and A-OKVQA, we observe a substantial drop in RAG workflow performance compared to direct answering. This is because the majority of questions in these datasets ($> 50\%$) can be answered without search, and incorporating web-search results into the model's reasoning introduces noise, leading to a performance decline. In contrast, the prompt-based search agent baselines exhibit a more stable performance gain, as the model is explicitly prompted to invoke multimodal search tools and incorporate retrieved results only when necessary. However, since these models are not explicitly trained to interact with web-search tools, their performance remains inferior to that of web-search equipped MLLMs. DeepMMSearch-R1-7B (RL) delivers the largest performance

| Model | Average | InfoSeek | Enc-VQA | SimpleVQA | DynVQA | OKVQA | A-OKVQA |
|---|---|---|---|---|---|---|---|
| *Direct Answer* | | | | | | | |
| InternVL2.5-8B | 40.46 | 17.57 | 19.70 | 35.44 | 13.71 | 74.61 | 81.75 |
| InternVL3-8B | 41.53 | 16.85 | 21.50 | 37.51 | 17.38 | 72.85 | 83.06 |
| Qwen-2.5VL-7B | 43.11 | 26.38 | 18.75 | 47.48 | 20.14 | 63.10 | 82.79 |
| Qwen-2.5VL-32B | 50.04 | 31.09 | 27.25 | 47.29 | 29.23 | 78.22 | 87.16 |
| GPT-4o | 50.16 | 35.96 | 27.15 | 48.27 | 31.19 | 71.96 | 86.46 |
| o3 | 60.38 | 48.22 | 49.15 | 53.11 | 41.68 | 80.36 | 89.78 |
| *RAG Workflow* | | | | | | | |
| Qwen-2.5VL-7B | 36.00 | 41.13 | 38.95 | 29.71 | 39.02 | 34.64 | 32.58 |
| Qwen-2.5VL-32B | 35.50 | 40.20 | 42.00 | 28.53 | 40.98 | 35.37 | 25.94 |
| GPT-4o | 41.50 | 45.86 | 44.50 | 35.93 | 43.22 | 38.76 | 40.70 |
| o3 | 47.49 | 50.34 | 49.15 | 35.74 | 47.41 | 51.70 | 50.57 |
| *Prompt based Search Agent* | | | | | | | |
| Qwen-2.5VL-7B | 48.24 | 29.75 | 27.85 | 46.89 | 22.38 | 77.15 | 85.41 |
| Qwen-2.5VL-32B | 50.94 | 28.61 | 32.80 | 48.67 | 40.00 | 72.87 | 82.71 |
| *Web-Equipped MLLMs* | | | | | | | |
| MMSearch-R1-7B* | 50.56 | 41.33 | 36.85 | 53.90 | 40.14 | 59.89 | 71.27 |
| **DeepMMSearch-R1-7B (SFT)** | 56.23 | 47.45 | 50.35 | 52.02 | 43.08 | 67.52 | 76.94 |
| **DeepMMSearch-R1-7B (RL)** | 57.13 | 47.51 | 52.25 | 55.87 | 45.87 | 67.80 | 73.45 |

Table 1: Performance comparison of DeepMMSearch-R1 with baselines of three categories. *The reported numbers are obtained by evaluating the model using the same image-search and text-search APIs that we use. For a fair comparison, we follow the evaluation prompt on which the baseline was trained.

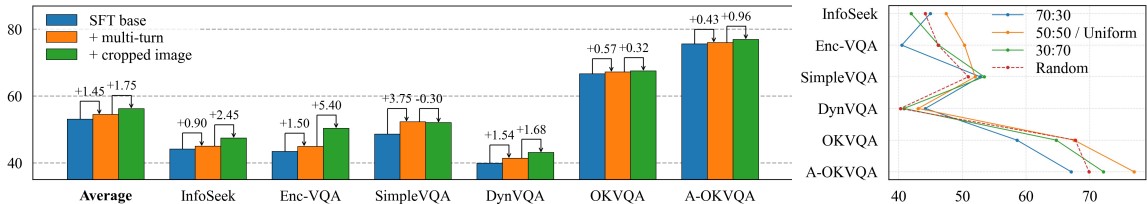

Figure 3: **(left)** Impact of self-reflection, self-correction and cropped image search on performance. **(right)** Effect of the ratio of search-required to search-free data, and of sampling strategies when curating SFT data.

boost, validating the importance of training models to use search tools rather than relying on fixed retrieval strategies or test-time prompting. These results validate that fine-tuning models to leverage search tools and associated tag schema improves performance and also makes the retrieval cost effective by making web-search more efficient and intelligent.

**Cropped image search and distilled self-reflection and self-correction capabilities boost performance.** We showcase the impact of enabling multiple text searches and cropped image search capability in Figure 3(left).The SFT base model refers to the setup with whole-image search and a single text search call. We see that, on average, performance improves with distilled self-reflection and self-correction. This enables the model to iteratively refine its queries in response to retrieved web results and better navigate noisy real-world information. A similar trend is observed with cropped image search, yielding an average improvement of $+1.75$ across six datasets, highlighting its effectiveness. It helps mitigate background noise and makes the search more effective, and is particularly useful for answering questions that concern a single visual entity in the image rather than the entire scene. We also observe that improvements on SimpleVQA and DynVQA are relatively higher, which aligns with expectations since these datasets are newer and contain a higher proportion of questions that require search.

**Search-balanced SFT data with examples uniformly sampled from all knowledge taxonomy categories provides better performance.** Firstly, we perform ablations with different ratios of search-required and search-free examples in the SFT data to study their impact on performance. From Figure 3(right), we observe that when the percentage of search-required questions is high, the fine-tuned model exhibits excessive search behavior and performs poorly on OKVQA and A-OKVQA, which require fewer search calls. Conversely, when the proportion of search-required questions in the SFT data is reduced, the model shows a performance drop on datasets with more challenging information-seeking questions,

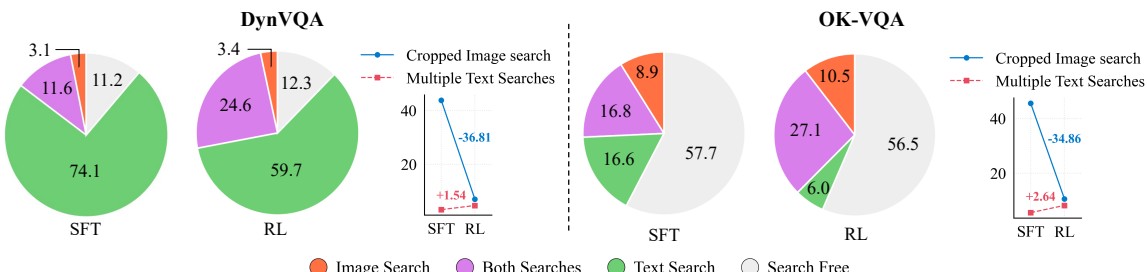

Figure 4: Tool usage statistics after SFT and RL on DynVQA and OK-VQA benchmarks.

such as InfoSeek, Enc-VQA, DynVQA, and SimpleVQA. We therefore conclude that a 50:50 balance provides the most effective configuration, as it distills search tool call behavior well and yields the best average performance across all datasets. Secondly, we examine the influence of maintaining a balanced distribution of examples across all categories in the knowledge taxonomy. As shown in Figure 3(right), uniformly sampling examples from each category leads to better average performance compared to random sampling.

**SFT enables tool use, while RL refines the tool-selection behavior by reducing unnecessary calls.** We summarized the tool usage of our model after the SFT and RL stages for two datasets in Figure 4. DynVQA is a newer dataset with more questions requiring external information, while OKVQA requires fewer search calls. The tool usage behavior of our model aligns with the nature of each dataset, leveraging tools for 87.7% of the samples in DynVQA compared to 43.5% in OKVQA.

Moving from the SFT to RL stage, we make three critical observations regarding tool use behaviour. (1) The model performs more image searches compared to text searches, resulting in an increase in both image search and mixed search tool calls. This behavior is desirable, as most questions are multimodal in nature, requiring both the identification of visual entities and the retrieval of factual information from the web. (2) After RL training, the model invokes multiple text searches more frequently, highlighting the role of RL in promoting self-reflection and self-correction. Specifically, we observe +1.54% and +2.64% more samples where the model refines its queries when the retrieved web information is insufficient to answer the question. (3) The model drastically reduces its reliance on cropped image searches (−36.81% on DynVQA and −34.86% on OK-VQA), yet still achieves overall performance gains. While this may seem counterintuitive, closer analysis shows that the model becomes more selective, and performs cropping operation only when necessary. For instance, the SFT model sometimes performed cropped image searches unnecessarily (examples provided in Appendix F.2), whereas the RL model corrected these errors. This observation reinforces the importance of RL in refining tool-use behavior and making it more efficient.

We further observe that DeepMMSearch-R1-7B (RL) exhibits cropped image searches or self-reflection behavior in 11.64% of questions on DynVQA and 18.95% on the OKVQA dataset, which constitutes a key part of our contributions. Overall, this analysis reinforces that SFT training equips the model with tool-use capabilities, while RL training refines tool selection, making it more efficient and better targeted for multimodal information-seeking tasks.

## 5 CONCLUSION

We propose DeepMMSearch-R1, a novel multimodal large language model designed to enhance visual question answering in knowledge-intensive and information-seeking contexts by integrating on-demand, multi-turn web search capabilities. Our approach addresses the limitations of prior retrieval-augmented methods and multimodal agents by enabling dynamic, iterative query refinement through self-reflection and self-correction, as well as incorporating a cropped image search tool. We achieve this with a two-stage training pipeline: (1) a supervised fine-tuning (SFT) stage using the proposed DeepMMSearchVQA, which equips the model with tool-use capabilities, followed by (2) online reinforcement learning (RL) via GRPO, which further refines tool-use behavior to make it more efficient. DeepMMSearch-R1 outperforms prior baselines across six benchmarks. We believe DeepMMSearch-R1 represents a compelling step forward in real-world, multimodal information-seeking AI, with promising applications in web agents, education, and digital assistance. **Future works** may explore expanding tool diversity, long-context reasoning, and scaling training to broader multilingual and multimodal domains.

## ETHICS STATEMENT

This work introduces methods to enhance multimodal LLMs with real-time web-search capabilities. While such systems offer clear benefits in improving informativeness and adaptability, they also raise ethical risks. Retrieved content may include biased, outdated, or misleading information, and automatic summarization can amplify misinformation or raise copyright concerns. Moreover, the approach depends on external infrastructure, which may limit accessibility for resource-constrained institutions. We encourage responsible deployment practices, including source attribution, content filtering, and human oversight in high-stakes applications.

## REPRODUCIBILITY STATEMENT

We have taken care to ensure that our work can be reproduced by the research community. All details of our training and evaluation setup are provided in the paper, including data generation pipeline, base model architecture, datasets, and training procedures. We report all hyperparameters used for both supervised fine-tuning and reinforcement learning, along with implementation details such as batch sizes, learning rates, and optimization schedules. Additionally, we provide the full prompts used for dataset generation, evaluation, and reward modeling in Appendix E. Together, these resources make it possible to replicate our experiments and verify our results.

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

## APPENDIX

In the appendix, we provide our LLM usage statement, discuss related works and mention the limitations and broader impact of our work. Additionally, we focus on its implementation and provide extensive details about the prompts used for dataset curation and the evaluation. Furthermore, we expand on the results presented in the main paper, providing general VQA experiments and some additional analysis. In the end, we provide visual examples of the proposed dataset.

## Table of Contents

## A LLM Usage Statement

We use large language model (LLM) as a supportive tool in this work. It was employed to assist in debugging portions of the code, generating and refining visualization figures, and improving the clarity of the manuscript through proofreading, grammar checking, and polishing the overall writing style. The LLM's role was limited to these supportive tasks, while all substantive research ideas, methodological decisions, analyses, interpretations, and final code implementations were developed and validated independently by us.

## B Related Works

### B.1 Multimodal LLMs

Multimodal large language models (MLLMs) combine visual encoders with powerful text-based large language models, enabling them to process and reason over both textual and visual inputs. Recent models such as GPT-4o Hurst et al. (2024), Gemini Team et al. (2023), Qwen2.5-VL Bai et al. (2025), InternVL Chen et al. (2024), LLaVA series Li et al. (2024a); Lin et al. (2023); Liu et al. (2023b;a; 2024a), Phi Series Abdin et al. (2024), Mantis Series Jiang et al. (2024a), OVIS series Lu et al. (2024b); Wang et al. (2025), VILA series Lin et al. (2024); Nath et al. (2025), Gemma series Team et al. (2024a;b) have demonstrated strong capabilities in visual perception, grounding, and multimodal reasoning, achieving remarkable progress in tasks like visual question answering, captioning, and multimodal dialogue. These advances highlight their potential as core components in real-world applications such as digital assistants, education, and information access. Despite these strengths, MLLMs face fundamental limitations in addressing knowledge-intensive or information-seeking queries Mensink et al. (2023); Chen et al. (2023); Cheng et al. (2025); Li et al. (2024c). Their training relies on static corpora, which inevitably leads to outdated knowledge as the real world evolves. Furthermore, the breadth of open-world knowledge follows a long-tail distribution, and it is infeasible to cover every rare or emerging fact within a fixed training dataset Mensink et al. (2023). This makes MLLMs struggle with long-tail knowledge and information that requires up-to-date context.

### B.2 RAG-based search

The RAG paradigm as the name suggests retrieves external information from a fixed knowledge corpora using vector search and augments it into the model context to generate factually grounded responses. Early contributions in this space include REALM Guu et al. (2020), which introduced retrieval-augmented pretraining by jointly optimizing a dense retriever with a language model to enable knowledge-intensive tasks. RAG Lewis et al. (2020) further advanced this paradigm by integrating a generative seq2seq model with neural retrieval, demonstrating strong gains in open-domain question answering. Recent efforts have extended retrieval augmentation to multimodal settings. REVEAL Hu et al. (2023) presented a retrieval-augmented visual-language pretraining framework, in which the memory, encoder, retriever and generator are all pre-trained end-to-end on a massive amount of data. VisRAG Yu et al. (2024) proposed a vision-language model based RAG pipeline that directly embeds documents as images for retrieval, avoiding information loss from text parsing. This strategy enables the joint filtering of retrieved documents, retaining only the most relevant and accurate references. RoRA-VLM Qi et al. (2024) introduced a two-stage retrieval process with image-anchored textual-query expansion to synergistically combine the visual and textual information in the query and retrieve the most relevant multimodal knowledge snippets. Moreover, they improve the robustness of retrieval-augmented vision-language model by injecting adversarial noise in the training process. RaR Liu et al. (2024e) proposed a retrieving-and-ranking augmented multimodal framework tailored for visual recognition, highlighting the role of retrieval quality in multimodal perception tasks. Recently, MMKB-RAG Ling et al. (2025) proposed a novel multi-modal RAG framework that leverages the inherent knowledge boundaries of models to dynamically generate semantic tags for the retrieval process. Despite these advances, RAG methods rely on static corpora, and work with an unrealistic assumption that all information can be captured within a fixed dataset. In real-world scenarios, web information is dynamic and constantly evolving, and the complexity of retrieval remains high. These factors pose significant challenges for adopting RAG in real-world, open-ended VQA.

                    16

### B.3 PROMPT-BASED SEARCH AGENTS

The prompt-based search agents act as plug-and-play modules that can be integrated with existing multimodal LLMs without additional finetuning. In this setup, the MLLM functions as an agent, incorporating web-retrieved information into its responses. For example, VSA Zhang et al. (2024) enables any vision-language model to operate as a multimodal automatic search engine. Its pipeline follows three steps: (1) visual content formulation, where the model identifies the object of interest; (2) web-knowledge search, where it generates multiple sub-questions and queries the web; and (3) summarization, where it consolidates the retrieved information before answering the user's query. Similarly, MM-Search Jiang et al. (2024b) introduces the MMSearch-Engine, a pipeline that augments large multimodal models with search capabilities through requerying, reranking, and summarization. OmniSearch Li et al. (2024c) further advances this idea by proposing a self-adaptive planning agent for multimodal retrieval. It dynamically decomposes complex questions into sequential sub-questions and selects retrieval actions accordingly. At each step, the planner evaluates prior retrieval feedback (via a solver) to decide whether to refine the query, switch retrieval mode (e.g., text, image, web), or generate new sub-questions. This flexible, feedback-driven process replaces rigid heuristics with a query-planning loop, better suited for dynamic, multi-hop, and multimodal VQA scenarios. However, across these approaches, the base model itself is not trained to engage effectively with web-retrieved information and external search tools, leaving it less capable of handling the noisy and complex nature of such real-world web information.

### B.4 WEB-SEARCH EQUIPPED MLLMs

Recent work focuses on R1-optimization of MLLMs to equip web-search capabilities in MLLMs. This trend follows from the success of reasoning models such as OpenAI o1,o3 and DeepSeek-R1. DeepResearcher Zheng et al. (2025) uses a multi-agent browsing architecture and the GRPO algorithm to learn to navigate, extract, and filter information from arbitrary web pages under realistic constraints (e.g., API limits, network latency, anti-crawling). R1-Searcher Song et al. (2025) presents a two-stage outcome-based reinforcement learning framework that allows LLMs to autonomously invoke external search systems during reasoning for knowledge-intensive tasks. In stage one, the model is rewarded for learning to trigger retrieval (without regard to answer correctness), and in stage two it is further trained to integrate retrieved evidence to maximize answer accuracy. Search-R1 Jin et al. (2025) incorporates retrieved-token masking, which prevents the RL objective from directly optimizing over retrieved content, stabilizing training when mixing generated and retrieved tokens. However, all these works are restricted to text search and are unable to perform an image search, which limits their applicability in mulitmodal knowledge-intensive question answering. MMSearch-R1 Wu et al. (2025) is the only prior work that performs multimodal retrieval, but it has notable limitations. First, although the model can autonomously decide which tool to use, it is constrained to a single invocation per tool, which limits its ability to revise decisions through self-reflection and self-correction. Second, in knowledge-intensive VQA tasks, it is essential to precisely identify the visual entity in the image that the question refers to. However, in real-world settings, background clutter and the presence of irrelevant visual entities often introduce noise into the retrieval process. This noise can hinder accurate localization of the target entity, leading to suboptimal retrieval and reduced effectiveness of image search in practice. To address these limitations, we propose DeepMMSearch-R1, which performs image search using relevant image crops and can iteratively refine its text search queries to better navigate noisy real-world web information.

## C DATASETS

### C.1 INFOSEEK

InfoSeek Chen et al. (2023) is a large-scale knowledge-intensive visual question answering dataset designed for information-seeking tasks. It consists of 8,900 human-written question–answer pairs over 806 entities and 527 entity types, as well as 1.35 million automatically generated QA triplets covering 11,481 entities across 2,739 entity types. The dataset is split into UNSEEN ENTITY and UNSEEN QUESTION partitions to test generalization. InfoSeek is widely used for evaluating multimodal models in knowledge retrieval and reasoning beyond surface-level recognition.

### C.2 FVQA

FVQA Wu et al. (2025) is a multimodal search VQA dataset constructed to enable evaluation and training of models that must decide when and how to perform external searches in a knowledge-intensive setting. The FVQA training split

(FVQA-train) comprises around 6,000 image–question–answer samples ("FVQA-auto-vc") focused on visual knowledge, plus 7,000 text-knowledge examples drawn from InfoSeek ("FVQA-auto-txt"), and an additional 800 manually annotated "FVQA-manual-train" samples. The test split (FVQA-test) is manually curated for higher quality and diverse knowledge demands.

## C.3 ENCYCLOPEDIC VQA

Encyclopedic VQA Mensink et al. (2023) is a large-scale visual question answering dataset that focuses on visual questions about detailed properties of fine-grained object categories and specific instances. It comprises 221,000 unique question–answer pairs, each associated with up to 5 different images, yielding a total of around 1,000,000 (1 M) image-question-answer instances. The dataset is backed by a controlled knowledge base derived from Wikipedia, where each QA is linked to supporting evidence from Wikipedia articles.

## C.4 SIMPLEVQA

SimpleVQA Cheng et al. (2025) is a multimodal benchmark created to evaluate the factuality of MLLMs in answering short, natural-language visual questions. It contains 2,025 high-precision image–question–answer pairs, spanning 9 task categories (e.g. Object Identification & Recognition, Time & Event, Person & Emotion, Location & Building, Text Processing, Quantity & Position, Art & Culture, Object Attributes) and 9 topic domains. SimpleVQA's design ensures coverage across domains, concise and clear answer formats, and suitability for automated evaluation (e.g. via LLM-as-judge). It is intended to challenge MLLMs' abilities to ground answers in factual knowledge rather than hallucinate, and is often used to probe the knowledge boundaries of vision-language models.

## C.5 DYNVQA

DynVQA Li et al. (2024c) is a benchmark dataset constructed to assess multimodal retrieval-augmented generation (mRAG) systems on dynamic visual question answering tasks that require adaptive retrieval strategies. It contains 1,452 questions spanning 9 domains, evenly split across English (715) and Chinese (737) items. The questions are categorized into three dynamic types: (1) those with rapidly changing answers (385 questions, ∼26.5%), (2) multimodal-knowledge questions requiring non-textual evidence (178 questions, ∼12.3%), and (3) multi-hop questions requiring multi-step reasoning (112 questions, ∼7.7%). Across all questions, 59.6% require external visual knowledge beyond what is directly in the image, and 26.7% require more than two reasoning hops. DynVQA is designed with temporal dynamism, and some answers may change over time. Therefore, the dataset is periodically updated to maintain answer correctness.

## C.6 OKVQA

OKVQA Marino et al. (2019) is a knowledge-based visual question answering dataset in which the visual content alone is insufficient to answer questions—models must draw on external knowledge. It comprises 14,055 open-ended question–answer (QA) pairs associated with 14,031 images. Each QA is annotated with 5 ground truth answers per question. To reduce dataset bias, frequently repeated answers were pruned, such that questions whose most common answer appeared more than 5 times were removed. The dataset covers a diverse set of 10 knowledge categories (e.g., Vehicles & Transportation; Cooking & Food; Science & Technology) determined via crowd annotations. Baseline VQA models that perform well on standard VQA benchmarks show significant performance drops on OKVQA, highlighting the difficulty of knowledge retrieval and reasoning in this setup.

## C.7 A-OKVQA

A-OKVQA (Augmented OK-VQA) is a crowdsourced visual question answering benchmark designed to require commonsense and world knowledge beyond simple fact lookup. It comprises approximately 24,903 question–answer–rationale triplets spread across 17.1K training, 1.1K validation, and 6.7K test splits. Each question is accompanied by both multiple-choice (MC) options and direct-answer (DA) alternatives, along with a rationale (one explanatory sentence) for the train split. To ensure diversity, A-OKVQA filters out trivial or overly repetitive questions, enforces quality control via crowd annotation, and clusters similar images to discourage repetitive phrasing. Compared

to OKVQA, A-OKVQA contains about twice as many questions and adds rationale annotations to support explainable reasoning.

## D IMPLEMENTATION DETAILS

We use the LLaMA-Factory framework to perform supervised finetuning. Our base model is Qwen2.5-VL-7B-Instruct, which we finetune using LoRA with a rank of 8 applied across all target modules. Training is performed for 3 epochs with a learning rate of 1e-4, following a cosine scheduler with a warmup ratio of 0.1. We enable bf16 mixed precision for computational efficiency. The model is trained using 1 node with 8 Nvidia H100 GPUs, with per-device batch size is set to 1, with gradient accumulation of 1 step, resulting in a global batch size of 8. Since the VQA dataset consists of multi-turn conversations, we apply input masking during training to ensure that the model is optimized only on the generated outputs. For online RL optimization, we adopt the GRPO algorithm implemented in the veRL framework. The reward model is GPT-4o, which evaluates generated responses and provides feedback for optimization, with $\lambda_{\text{fmt}}$ set to 0.1. We apply a KL-penalty with a coefficient of 0.001 and the clip ratio is set to 0.2. Training is performed for 20 epochs on 4 nodes each with 8 Nvidia H100 GPUs. We use a batch size of 256 with a mini-batch size of 64, and set the rollout number to 8 per iteration. A warmup phase of 45 steps is applied to stabilize learning rates, which are initialized at 2e-6. The image search/cropped image search tool can be called once while the text-search tool can be called multiple times, with total tool calls restricted to 10 per rollout. The maximum response length is set to 8192 tokens We again mask the input tokens to ensure that optimization focuses only on the generated outputs.

                 19

# E PROMPTS

## E.1 SFT DATASET GENERATION

---

**Initial Prompt**

You are an expert visual assistant. Your task is to answer a user's question based on the provided image.

**Step 1: Analyze the Image**

Carefully examine the image and the user's question: `{question}`. Identify all recognizable entities, objects, text, and other visual clues.

**Step 2: Plan Your Action**

Based on your analysis, you must perform one of the following actions. You must include your thinking process inside a `<reason>...</reason>` block before choosing an action.

- **Action 1: Answer Directly**

  If you can confidently identify the visual element and have the internal knowledge regarding the facts sufficient to answer the question, provide a direct, concise answer inside `<answer>...</answer>` tag.

  Example: `<answer>The construction of Eiffel Tower was finished on March 31, 1889.</answer>`

- **Action 2: Use Image Search**

  If you are not sure about the visual element and need to identify the visual element in the image, you can use one of the following image search methods.

  - **Cropped search (Preferred for specific questions):** Use this if the question is clearly about a specific visual element such as an object, person, animal, plant, aircraft, etc., or if the background is irrelevant. Describe the visual element concisely inside the `<img_search>...</img_search>` tags.

    Example:
    `<img_search>the face of the person on the left</img_search>`
    `<img_search>the red logo on the baseball cap</img_search>`

  - **Whole image search:** Only use this if the question is about the entire scene in general, its location, or the overall context. Output only: `<img_search></img_search>`.

    **Note:** Do not output `<img_search></img></img_search>`.

- **Action 3: Use Text Search**

  If you can identify the visual element confidently but need more specific information to answer the question, invoke the text search tool. Generate a focused query and output it as `<text_search>your search query</text_search>`.

Remember, search results will be provided to you in subsequent turn. You can analyze the search results and decide your next action. You can perform image search only once, but have the option to perform multiple text searches to gather relevant information. All search results will be placed inside `<information>...</information>`.

Here is the image and question: `<image>{question}`

---

**After Image Search**

You have received information from an image search. Your goal is to use this new information to answer the original question: {question}.
**Step 1: Analyze the Results**
Review the provided information within the `<information>...</information>` block. Synthesize what you've learned about the visual element in question.
**Step 2: Plan Your Next Action**
Include your thinking process inside a `<reason>...</reason>` block. Then, choose one of the following actions:

- **Action 1: Answer Directly**
  If the image search results have helped you identify the visual element and you can confidently answer the question with your internal knowledge, provide the final, concise answer inside an `<answer>...</answer>` tag.

- **Action 2: Use Text Search**
  If the image search results have helped you identify the visual element but you need more specific details to answer the question, invoke the text search tool. Formulate a precise query based on the image search results and output it as `<text_search>your search query</text_search>`. You can use the text search tool multiple times in subsequent turns if needed.

**After Text Search**

You have received results from a text search. Your goal is to analyze this new information and decide the next best step to answer the original question: {question}.
**Step 1: Analyze the Results**
Review the new information provided in the `<information>...</information>` block. Compare it against the information you already have and what is still needed to answer the question.
**Step 2: Plan Your Next Action**
Include your thinking process inside a `<reason>...</reason>` block. Then, choose one of the following actions:

- **Action 1: Answer Directly**
  If you have now gathered all the necessary information, provide the final, concise answer inside an `<answer>...</answer>` tag.

- **Action 2: Search Again**
  If the results are helpful but still insufficient, perform another text search. Create a new, more specific, or modified query to find the missing facts. Output the new query as `<text_search>your refined search query</text_search>`.

- **Action 3: Give Up**
  If you have exhausted your search attempts and believe the answer cannot be found from the provided information, conclude by outputting `<answer>Unable to answer due to lack of relevant information</answer>`.

E.2 WEBSEARCH EQUIPPED MLLMS EVALUATION PROMPT

The evaluation prompt is same as SFT data generation prompts as detailed in Section E.1.

## E.3 RAG WORKFLOW PROMPT

---
**Initial Prompt**

---

You are a helpful assistant designed to answer questions about images using external knowledge. You are given a question accompanied by an image that cannot be answered without external knowledge.

You are provided with a question, the corresponding image, and a text summary from a reverse image search that identifies the main visual subject. Based on all this information, your task is to formulate a single, effective query for a search engine (e.g., Google) to find the specific facts needed to answer the question.

**Question**: {question}

**Reverse Image Search Information**: {information}

Provide only the text query you will use for the search, in the format `<text_search>your query</text_search>`.

---
**Final Answer Prompt**

---

You have now received the results from your text search. Your goal is to analyze the text search results to provide a final concise answer to the original question based on the image provided.

**Original Question**: {question}

**Text Search Results**: {information}

**Follow the following process:**

1. Briefly explain your reasoning process by analyzing the facts from the search results that are relevant to the question. Enclose this reasoning inside `<reason>your reason</reason>` tags.

2. Provide the final, direct answer to the question between `<answer>` and `</answer>` tags. If the information is insufficient, respond **ONLY** with:
   `<answer>Unable to answer due to lack of relevant information.</answer>`

## E.4 PROMPT-BASED SEARCH AGENT PROMPT

The prompt-based search agent prompts are same as SFT data generation prompts as detailed in Section E.1.

## E.5 LLM-AS-JUDGE PROMPT

---
**LLM-as-judge Prompt**

---

You are an impartial judge evaluating a model's answer for a visual question answering task. Your task is to determine if the **Predicted Answer** is correct by comparing it against the **Ground-Truth Answer(s)**.

**IMPORTANT INSTRUCTION:** The **Ground-Truth Answer(s)** field may contain alternate correct answers. The predicted answer should be considered **CORRECT** if it is semantically equivalent to **at least ONE** of the provided ground-truth answers.

Please respond with only [CORRECT] if the prediction is correct, and [INCORRECT] otherwise.

**— Evaluation Details —**
**Question**: {question}
**Ground-Truth Answer(s)**: {references_for_prompt}
**Predicted Answer**: {candidate}

                 22

## E.6  GPT AS REWARD MODEL PROMPT

---

**GPT-4o as reward model prompt**

You are a strict evaluation judge for short-answer matching. Given a model's final answer and a list of gold answers, decide if the model's answer matches **ANY** gold answer.
**Rules:**

1. **Semantic Equivalence:** Consider synonyms, paraphrases, and common aliases as valid matches.
   Example: `"NYC"` ≈ `"New York City"`.

2. **Ignore Trivial Differences:** Do not penalize differences in articles, punctuation, word order, or casing.
   Example: `"The Pacific Ocean"` ≈ `"pacific ocean"`.

3. **At Least One Match:** If the model's answer aligns with ANY gold answer based on the rules, set `match=true`. Otherwise, `match=false`.

4. **Numerical Flexibility:** For answers involving numbers, an answer is a MATCH if it meets any of these criteria:
   (a) **Range Inclusion:** The model provides a range that contains the gold answer.
       Example: Model: `"20 to 24"`, Gold: `["21"]`.
   (b) **Reasonable Rounding:** The model's answer is a reasonably rounded version of the gold answer.
       Example: Model: `"176"`, Gold: `["176.124"]`.
   (c) **Unit Conversion:** The model's answer is equivalent but in a different unit.
       Example: Model: `"3 km"`, Gold: `["3000 m"]`.

5. **Substantive Difference:** If the meaning, entity, or value differs in a way not covered by the rules above, it is NOT a match.
   Example: `"Jupiter"` ≠ `"Mars"`.
   Example: `"5.2"` ≠ `"52"`.
   Example: Model: `"10-15"`, Gold: `["16"]` → **NO MATCH**.

**Output Format:**
```
MATCH: true/false
REASON: A concise explanation focusing only on why the answer matches or
does not match.
```

---

## E.7  LLM SUMMARIZER PROMPT

---

**Image Search Summarization Prompt**

Based on the following text extracted from the title and description of the retrieved images obtained from a Google Lens search, concisely describe the primary visual content (such as faces, objects, locations, events, logos, or text) of the original image in four to five sentences.
**Extracted Text:**
```
{formatted_results}
```

---

          23

**Text Search Summarization Prompt**

Review all the provided text references to find the most relevant information to answer the question. Analyze the relevant facts from these references into a single, concise summary of 10–12 sentences that answers the question.
**Question**: {original_question}

**References:**
{references_text}

# F  ADDITIONAL CONTENT AND RESULTS

## F.1  INTRODUCTION FIGURE

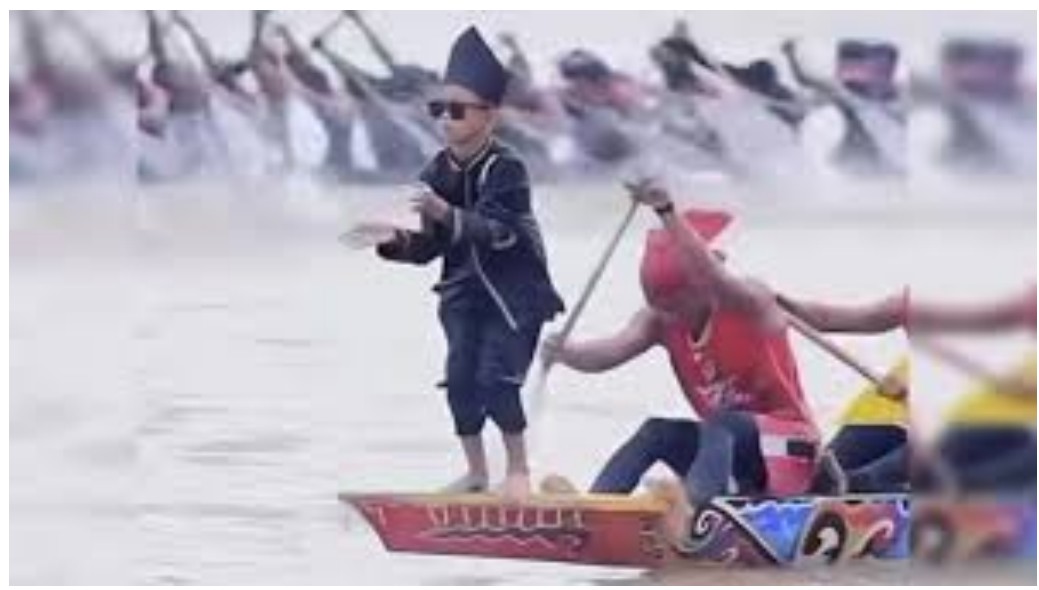

Figure F.1: An image of a boat race.

## F.2 SFT MODEL CROPPING EXAMPLE

We present examples where the SFT model performs unnecessary cropping in Figure F.2. RL training with GRPO corrects this issue, making tool usage more efficient. The RL-optimized model performs cropping only when required.

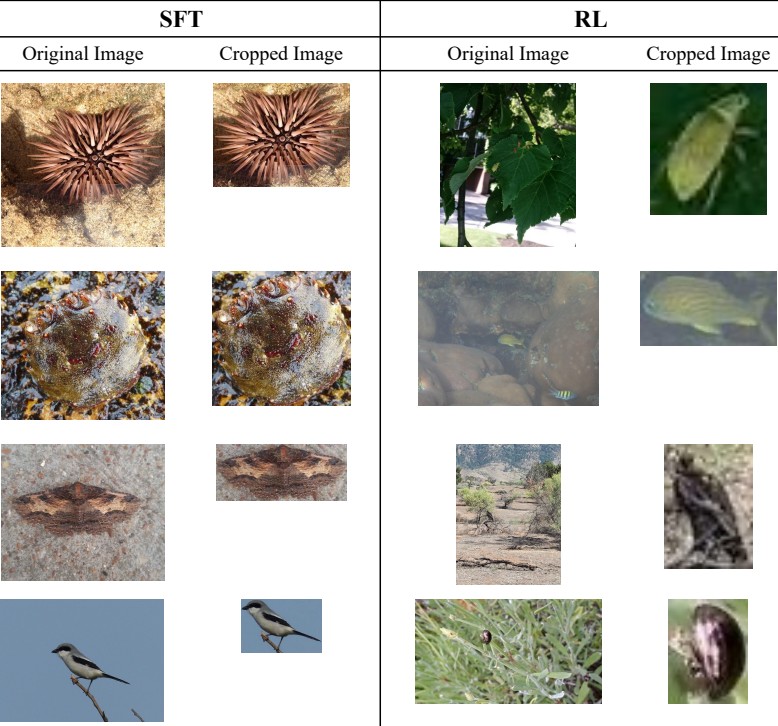

Figure F.2: Croppd image search usage: SFT model vs RL optimized model

## F.3 PERFORMANCE ON GENERAL VQA

To quantify the impact of SFT + RL training on the proposed model's general VQA and reasoning ability, we benchmark DeepMMSearchR1-7B (RL) on a range of benchmarks, including OCRBench Liu et al. (2024d), MMVet Yu et al. (2023), AI2D Kembhavi et al. (2016), MathVista Lu et al. (2024a), MMBench Liu et al. (2024c), DocVQA Mathew et al. (2021), and InfoVQA Mathew et al. (2022). We observe that the model maintains its overall performance while achieving improvements on MathVista and MMVet. These results suggest that the proposed model effectively learns to interact with web-search tools while preserving its general visual understanding and reasoning capabilities.

| Models | OCRBench | MMVet | AI2D | MathVista MINI | MMBench | DocVQA | InfoVQA |
|---|---|---|---|---|---|---|---|
| Qwen2.5-VL-7B-Instruct | 88.30 | 68.30 | 83.74 | 68.20 | 83.84 | 94.97 | 82.58 |
| DeepMMSearch-R1-7B (RL) | 87.60 | 69.81 | 82.57 | 66.80 | 83.76 | 94.63 | 81.63 |

Table F.1: Comparison of model performance on GeneralVQA.

## G  DEEPMMSEARCHVQA SAMPLES

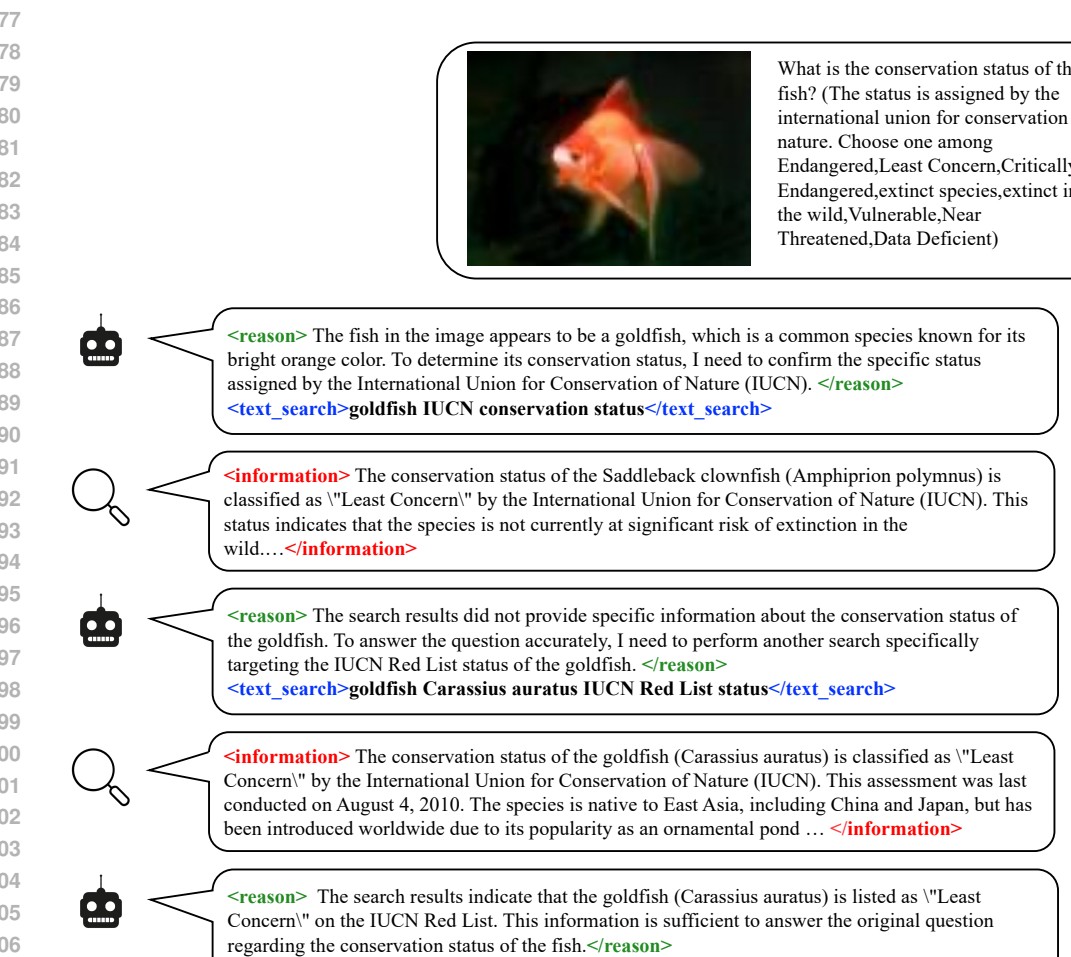

Figure G.1: A sample in DeepMMSearchVQA

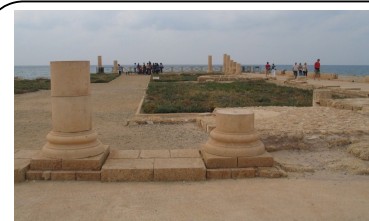

What country does this city belong to?

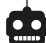

**<reason>** The user is asking for the country where the city in the image is located. The image shows ancient ruins, including columns and building foundations, next to the sea. The style of the ruins is reminiscent of Roman or Greek architecture, which is common in many countries around the Mediterranean. To accurately identify the specific location and thus the country, I need to perform an image search. A whole image search is suitable as the entire context of the ruins and the coastline is important for identification.**</reason>**
**<img_search></img_search>**

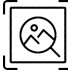

**<information>** Based on the extracted text, the original image most likely features the archaeological ruins at Caesarea National Park in Israel. The scene is set on the Mediterranean coast, showing ancient stone structures from the Roman or medieval periods. These ruins are part of a significant historical site that was once an ancient port city. The photograph may show a wide view of the park against the sea or a close-up of a specific artifact, such as the inscribed Pilate Stone.**</information>**

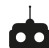

**<reason>** The image search results have identified the location in the image as the archaeological ruins at Caesarea National Park. The information explicitly states that this park is in Israel. Therefore, I have sufficient information to directly answer the user's question about the country.**</reason>**
**<answer>The city shown in the image is Caesarea, and it is located in Israel.</answer>**

Figure G.2: A sample in DeepMMSearchVQA

# H LIMITATION AND BROADER IMPACT

## H.1 LIMITATION

Despite the demonstrated benefits, our work faced several limitations. First, reliance on multiple search tools inherently increases susceptibility to errors arising from tool failures, latency, API query limits, or scraping blocks, all of which can disrupt the reasoning process. Secondly, since web-search tools are dynamic and continuously updated, retrieval outcomes may vary over time, introducing variability in both training and evaluation.

A major bottleneck we encountered during training was performing online GRPO with live web-search tools, which posed challenges in terms of stability and reliability. To mitigate this, we had to implement extensive fail-checks, retries, and safeguards to ensure robust information extraction from the web tools. These limitations highlight that integrating real-time web-search retrieval is not a trivial task and requires substantial efforts.

## H.2 BROADER IMPACT

Our work has the potential to significantly advance multimodal information-seeking systems by enabling multimodal LLMs to dynamically retrieve and reason over real-world knowledge. This opens up promising applications in education, digital assistants, research support, and accessibility tools, where timely and accurate information retrieval is essential.

However, reliance on web-search also introduces risks, including amplification of misinformation, propagation of biased or low-quality sources, and challenges related to copyright when models retrieve or summarize content from proprietary sources. Furthermore, depending on poorly reliable web data can undermine factual accuracy and erode user trust. To address these concerns, future work should prioritize incorporating mechanisms for source attribution and quality filtering of retrieved content. We emphasize the importance of responsible deployment, with safeguards to ensure factual reliability, and equitable access, so that such systems can be harnessed in a safe and beneficial manner.

