# OpenReview forum: "DeepMMSearch-R1: Empowering Multimodal LLMs in Multi-Modal Web Search"
_ICLR.cc/2026/Conference — ICLR 2026 Conference Withdrawn Submission_

### Official Review · Reviewer_bfx2 · 2025-10-28

**Soundness:** 3
**Presentation:** 2
**Contribution:** 1
**Rating:** 2
**Confidence:** 3

**Summary:**

This paper introduces DeepMMSearch-R1, a multimodal large language model designed to address multi-turn web searches for knowledge-intensive visual question answering. The system integrates three tools, text search, image search, and GroundingDINO for image cropping. The key innovation is enabling iteratively text query refinement (self-reflection and self-correction), and allowing the model to perform cropped image searches for specific visual entities. The training pipeline consists of supervised fine-tuning on a newly proposed DeepMMSearchVQA dataset with 10K samples, followed by online reinforcement learning using GRPO. The model demonstrates improvements over baselines on multiple benchmarks including InfoSeek, Encyclopedic-VQA, SimpleVQA, and DynVQA.

**Strengths:**

< Strength >

- This paper tries to bridge multimodal retrieval with iterative planning, which is timely and practically important.
- Extension with cropped image search by integrating GroundingDINO as an intermediate cropping tool addresses a practical challenge in multimodal search.
- The DeepMMSearchVQA dataset with structured tool-call annotations could be valuable for future research for tool-use behaviors in multimodal models
- The paper presents concrete ablation studies examining key components: multi-turn search capability, cropped image search, and data distribution effects (e.g. Figure 3)
- The prompts and training details for SFT and RL are well described

**Weaknesses:**

< Weakness >

- The paper is primarily an extension of Search-R1 [1] to the multimodal domain and the incremental nature of the contribution diminishes its impact for a top-tier venue. Further, this work essentially applies existing methods, GroundingDINO to multimodal settings, without proposing novel methods to multimodal-specific challenges. The web-search equipped MLLM is also studied in other papers, such as [2, 3].
- There is an unexplained design choice. Image search is restricted to a single use while text search allows multiple iterations. This asymmetry is not justified despite the fact that it meaningfully limit the coverage of the proposed method.
- The paper use gpt-5-chat-latest as a reward model even when ground truth is available. Although this can give more flexibility, this choice wastes computational resources. The marginal gain from RL over SFT (56.23 → 57.13, less than 1%) raises questions about whether the computational cost of RL training is justified. Although Lines 392-405 claim RL contributes to adapting data distribution for more image search and self-reflection, the final performance doesn't strongly support that this adaptation works well in practice
- The paper relies on an "in-house" image search API and web search API without describing their characteristics or providing alternatives. Although all baseline leverages same API, these omissions contradict the authors' claim about reproducibility and might affect to the final performance. Additional ablation comparing their APIs with well-known public retrieval methods (e.g., CLIP-based retrieval) would be needed.
- Given multiple tools and corresponding prompts, MMSearch-R1 would be practically the only baseline that can be compared apple to apple, yet the paper provides minimal description of its algorithm or how the proposed work improves upon it.
- As the pipeline leverages various API calls, analysis of computational costs would be desired but insufficient (latency, API calls, inference time).


< Minor issuses >

- Figure 4's format is confusing with three subfigures (SFT, RL, line graph) per category. A clearer presentation would show just SFT and RL side-by-side using unified category.
- Inconsistency between Figure 1 notation `<img_search>img</img_search>` and Appendix E.1 notation `<img_search><img></img_search>`
- Training objective should include precise annotation for the loss masking, for example, Search-R1 paper [1] introduced $I(y_t)$ to denote token loss masking operation in Equation 2.


[1] Jin, Bowen, et al. "Search-r1: Training llms to reason and leverage search engines with reinforcement learning." arXiv preprint arXiv:2503.09516 (2025).
[2] Wu, Jialong, et al. "Webwalker: Benchmarking llms in web traversal." arXiv preprint arXiv:2501.07572 (2025).
[3] Go, Dongyoung, et al. "CUE-M: Contextual Understanding and Enhanced Search with Multimodal Large Language Model." arXiv preprint arXiv:2411.12287 (2024).

**Questions:**

- The paper introduces DeepMMSearchVQA as one of its main contributions. Could the authors clarify whether they have plans to make this dataset publicly available?
- Regarding the choice of GPT-5 as the reward model, I'm curious about the rationale behind this decision. For semantic comparison with ground truth answers, would a smaller but capable model (such as Qwen-30B) potentially achieve similar performance while being more accessible to researchers?
- To enhance reproducibility, would the authors consider providing additional details about their search APIs or, alternatively, presenting additional results using publicly available retrieval methods (e.g. CLIP-based retrieval for images)? This would help the community better understand and replicate the work.

---

### Official Review · Reviewer_k28S · 2025-10-31

**Soundness:** 3
**Presentation:** 2
**Contribution:** 2
**Rating:** 4
**Confidence:** 4

**Summary:**

This paper introduces DeepMMSearch-R1, a multimodal LLM that enhances knowledge-intensive VQA through on-demand, multi-turn web search with dynamic text and cropped image queries. It proposes a novel dataset (DeepMMSearchVQA) and a two-stage training pipeline (SFT + GRPO) to teach the model when and how to search, achieving strong performance across benchmarks.

**Strengths:**

- DeepMMSearch-R1 can perform on-demand, multi-turn web searches with both text and image tools, enabling dynamic knowledge updates.

- The model introduces an intermediate cropping tool that automatically selects the most relevant image region, significantly reducing background noise in visual search.

- Extensive ablations and tool-usage statistics demonstrate that SFT equips the model with tool-use capabilities, while RL further refines tool selection for real-world deployment.

**Weaknesses:**

- Insufficient analysis, the work does not quantify how often image/text search tools fail or return noisy results, and provides no robustness evaluation against adversarial or empty tool responses.

- The RL stage uses the GPT-5 as the reward model, introducing reward-hacking risk without calibration or human-validation analysis (maybe use more metric for eval is better).

**Questions:**

- How is the performance of API-based models, such as Gemini 2.5 which is used to construct the data?

- The paper mentions that different data mixing ratios will bring different results. As different benchmarks will have different characteristics (some require more searching, some require more cropping, and some require more mathematical or logical thinking). How can using single InfoSeek as SFT qa pair guarantee all of these capabilities? Maybe more experiments is needed.

- The reward doesn't supervise the model's use of tools. Can using a single data source guarantee that the model effectively learns the appropriate parameter distribution (the model's ability to call tools)? Have you observed a phenomenon: the model increasingly tries to crop images or quickly stops, etc. ?

- More multi-turns case is needed.

---

### Official Review · Reviewer_UepG · 2025-11-01

**Soundness:** 2
**Presentation:** 3
**Contribution:** 2
**Rating:** 4
**Confidence:** 4

**Summary:**

This paper introduces DeepMMSearch-R1, a MLLM trained to perform multi-turn web searches for knowledge-intensive visual question answering. The key contributions include: (1) a novel training dataset containing 10K multi-turn conversations with search tool annotations and web-retrieved information, (2) a multimodal search pipeline integrating text search, image search, and a grounding tool for cropped image retrieval, and (3) a two-stage training approach combining SFT and RL. The model demonstrates the ability to perform self-reflection and self-correction by iteratively refining queries, and achieves competitive performance with o3 on knowledge-intensive VQA benchmarks.

**Strengths:**

1. The idea of using grounding tools to crop the relevant image part is novel and reasonable. The design of multi-turn and self-refining is also reasonable, especially in the search field.

2. The authors have done extensive work from data curation to the two-stage training method.

**Weaknesses:**

1. The primary concern is the unreasonable design of the image cropping tools, which lack a validation phase in two key aspects. First, the method fails to verify whether GroundingDINO accurately locates the target object. Second, even if GroundingDINO crops the correct region, the method does not confirm whether the images retrieved via the visual search API match this cropped region. For instance, the search API might return three correct and two incorrect images, yet only the summarized information of all five images is provided to the model, leading to serious confusion. Additionally, only text is delivered when optimizing image cropping and returning search results, which greatly impairs the reliability of the proposed technique.

2. When training with RL, the authors use GPT-5 to calculate the reward. It is questionable whether GPT-5 is necessary here, as it may substantially increase training costs and hinder reproducibility.

3. Insufficient comparison. MMSearch-R1 also incorporates results from MMSearch and LiveVQA. The authors are expected to provide the results of these benchmarks.

**Questions:**

1. How long does it take for the RL training stage? The concern is whether the search API could cause the training to be very slow.

2. Is the SFT phase necessary? What if the authors only use RL to train?

---

### Official Review · Reviewer_ziRc · 2025-11-01

**Soundness:** 3
**Presentation:** 3
**Contribution:** 3
**Rating:** 4
**Confidence:** 5

**Summary:**

This paper proposes DeepMMSearch-R1, an MLLM tailored for multimodal web-search question-answering. It introduces a curated training dataset and a two-stage RL-based tuning pipeline. Experimental results demonstrate strong performance across multiple benchmarks.

**Strengths:**

High-quality dataset. The authors release DeepMMSearchVQA, a valuable dataset that can benefit the community and inspire future research in multimodal web-search tasks.

Clear presentation. The paper is clearly written and well-organized, with intuitive figures that make the technical contributions easy to understand.

Comprehensive analysis. The ablation studies and experimental analyses are thorough and informative.

**Weaknesses:**

Limited evaluation scope. The current evaluation covers seven benchmarks, most of which are not strongly aligned with web-search scenarios. Given that MLLMs require broad evaluation to demonstrate generalization, additional benchmarks such as MathVista, MMMU, ZeroBench, and MMSearch are recommended. Including more diverse web-search-related tasks would further validate the model’s effectiveness.

Cropping-based search strategy. The proposed design of cropping image regions for search raises concerns. What happens when useful context exists outside the cropped region? How does the model ensure the selected region is optimal? Clarification on whether the RL reward explicitly supervises cropping decisions would strengthen this contribution.

**Questions:**

see weakness

---

### Note · Authors · 2025-11-13

I have read and agree with the venue's withdrawal policy on behalf of myself and my co-authors.